# A Novel Magnetic Resonance Imaging-Based Radiomics and Clinical Predictive Model for the Regrowth of Postoperative Residual Tumor in Non-Functioning Pituitary Neuroendocrine Tumor

**DOI:** 10.3390/medicina59091525

**Published:** 2023-08-23

**Authors:** Chaodong Shen, Xiaoyan Liu, Jinghao Jin, Cheng Han, Lihao Wu, Zerui Wu, Zhipeng Su, Xiaofang Chen

**Affiliations:** Department of Neurosurgery, First Affiliated Hospital of Wenzhou Medical University, Wenzhou 325000, China; shenchaodong123@163.com (C.S.); aliwaipo2@163.com (X.L.); jinjinghao101@163.com (J.J.); finncavendish@163.com (C.H.); 44259609@163.com (L.W.); doctorwzr@163.com (Z.W.)

**Keywords:** pituitary neoplasms, residual, risk factors, magnetic resonance imaging, nomograms

## Abstract

*Background and Objectives*: To develop a novel magnetic resonance imaging (MRI)-based radiomics–clinical risk stratification model to predict the regrowth of postoperative residual tumors in patients with non-functioning pituitary neuroendocrine tumors (NF-PitNETs). *Materials and Methods*: We retrospectively enrolled 114 patients diagnosed as NF-PitNET with postoperative residual tumors after the first operation, and the diameter of the tumors was greater than 10 mm. Univariate and multivariate analyses were conducted to identify independent clinical risk factors. We identified the optimal sequence to generate an appropriate radiomic score (Rscore) that combined pre- and postoperative radiomic features. Three models were established by logistic regression analysis that combined clinical risk factors and radiomic features (Model 1), single clinical risk factors (Model 2) and single radiomic features (Model 3). The models’ predictive performances were evaluated using receiver operator characteristic (ROC) curve analysis and area under curve (AUC) values. A nomogram was developed and evaluated using decision curve analysis. *Results*: Knosp classification and preoperative tumor volume doubling time (TVDT) were high-risk factors (*p* < 0.05) with odds ratios (ORs) of 2.255 and 0.173. T1WI&T1CE had a higher AUC value (0.954) and generated an Rscore. Ultimately, the AUC of Model 1 {0.929 [95% Confidence interval (CI), 0.865–0.993]} was superior to Model 2 [0.811 (95% CI, 0.704–0.918)] and Model 3 [0.844 (95% CI, 0.748–0.941)] in the training set, which were 0.882 (95% CI, 0.735–1.000), 0.834 (95% CI, 0.676–0.992) and 0.763 (95% CI, 0.569–0.958) in the test set, respectively. *Conclusions*: We trained a novel radiomics–clinical predictive model for identifying patients with NF-PitNETs at increased risk of postoperative residual tumor regrowth. This model may help optimize individualized and stratified clinical treatment decisions.

## 1. Introduction

Pituitary neuroendocrine tumors (PitNETs) constitute 10–20% of all primary brain tumors [1,2]. Non-functioning pituitary neuroendocrine tumors (NF-PitNETs) account for 14–54% of PitNETs. These tumors with a diameter greater than 10 mm are often associated with mass symptoms such as visual impairment, headache and varying degrees of hypopituitarism without abnormal increases in serum hormone levels [3,4]. Despite advances in drug therapies for many tumor types, surgery remains the primary treatment option for NF-PitNET. Surgery for NF-PitNET principally involves a transnasal trans-sphenoidal approach using an endoscope or microscope [5]. Unfortunately, only 40–50% of surgical cases achieve complete resection, and at least 10–20% of wholly resected tumors recur after 5–10 years. When a residual tumor is found after surgery, the regrowth rates at 5 and 10 years were 40% and 50%, respectively [6,7,8]. Our understanding of the natural history of postoperative residual NF-PitNET is incomplete. Therefore, we require a model for predicting postoperative residual tumor regrowth.

Previous studies attempted to identify factors related to tumor regrowth after the complete resection of PitNETs, including age, sex and body mass index (BMI) [8,9]. Other studies have shown that imaging features also help predict tumor regrowth (i.e., Knosp classification, Hardy classification, cystic transformation, preoperative volume, preoperative tumor volume doubling time (TVDT), postoperative residual volume and surgical resection ratio) [4,10,11,12,13,14]. Molecular markers Ki67, p53 and p27 are risk factors for PitNETs [10,15,16]. However, few studies have reported factors associated with postoperative residual tumor regrowth. In a recent study, three protein molecules―including p16, Wif1 and TGF-β―were identified and combined with age and gross tumor volume to build a model for predicting tumor regrowth [4]. Unfortunately, this study did not assess the risk of tumor regrowth relative to imaging features.

Radiomics is an emerging field that extracts the automatic quantification of imaging phenotypes to transform radiological images into mineable data [17]. Previous PitNETs radiomic studies have evaluated tumor subtypes, recurrence, consistency, prolactinoma resistance and cavernous sinus invasion [18,19,20,21,22,23]. However, to our knowledge, no study has used a radiomic approach to explore the risk of postoperative residual tumor regrowth in NF-PitNET.

This study sought to identify risk factors, including clinical and radiomic characteristics, that influence postoperative residual tumor regrowth. We constructed a model to predict postoperative residual tumor regrowth that may help to optimize treatment strategies for patients with postoperative residual tumors in NF-PitNET.

## 2. Materials and Methods

### 2.1. Patient Selection

This study was approved by the Clinical Research Ethics Committee of the First Affiliated Hospital of Wenzhou Medical University (permission number: 2022-527), and the requirement for informed consent was waived due to the retrospective study design. We retrospectively reviewed the data of patients with PitNETs treated at the local hospital from March 2013 to December 2019. We included (1) clinically NF-PitNET patients with postoperative tumor residue after the first operation, and the diameter of the tumors was greater than 10 mm; (2) with at least two years of documented clinical follow-up; and (3) patients with images performed using the same MRI equipment and protocol. We excluded (1) patients without adequate documentation of follow-up (clinical history); (2) patients without complete pre- and postoperative images using the desired MRI protocol; (3) patients treated with radiotherapy (RT) within two years of the first surgery; and (4) patients who underwent a second surgical approach within two years of the first surgery. A total of 325 patients clinically diagnosed with NF-PitNET underwent surgical treatment, and 135 patients with postoperative residual tumor were finally selected. We excluded 11 patients without complete follow-up data, 6 patients treated with radiotherapy within two years of the first surgery and 4 patients who underwent a second surgical approach within two years of the first surgery. Finally, we selected 114 patients with and without tumor regrowth. NF-PitNET regrowth was defined as an increase in maximum tumor diameter >2 mm from any direction on MRI beginning the day of surgery to the follow-up endpoint with or without the reappearance of visual disturbance, headache, or hypopituitarism [4]. The patient enrollment flowchart is shown in Figure 1.

### 2.2. Clinical Characteristics and Definitions

We collected the patients’ clinical, radiological and pathological characteristics, including sex, age, weight, height, BMI [BMI = weight (kg)/height^2^ (m^2^)], blood pressure, headache, vision changes, pituitary apoplexy, Knosp classification, cystic transformation, Hardy classification, consistency, preoperative TVDT, residual position, postoperative T1 enhancement, postoperative abnormal hormone level, postoperative diabetes insipidus, serum electrolyte index and ki-67 index.

We used computed tomography or MRI to confirm the diagnosis of pituitary apoplexy by revealing a pituitary tumor with hemorrhagic and/or necrotic components [24]. Cystic tumors were defined as tumors with >50% fluid content based on MRI T2 signal [13]. Patients with polyuria (urinary output > 300 mL/h for 3 h) and urine-specific gravity (USG) < 1.005 and at least one relative criterion (excessive thirst or serum sodium > 145 mmol/L) were considered to demonstrate postoperative diabetes insipidus [25]. TVDT (day), which was defined as the time it takes for the tumor volume to double, was calculated using the following formula: TVDT = (T2 − T1)*log2/log(V2/V1), where T1 is the previous imaging day and V1 is the volume and T2 is the subsequent day and V2 is the volume [10,26]. Tumor volumes were measured using the 3D-Slicer software (version 4.11). Both MRI examinations were completed preoperatively. Tumor consistency was jointly judged by two operating surgeons. In detail, tumors easily removable with maneuvers of suction were defined as soft, while those that were difficult to resection with a curette were classified as hard. If the suction of the majority of tumor was difficult, but resection with a curette was achievable, then consistency was evaluated as medium [15,27]. A ki-67 proliferative index > 3% was used to group patients [16,28]. During follow-up, residual tumor regrowth and other factors were strictly evaluated by two professional neurosurgeons in combination with a neuroradiologist and an endocrinologist.

### 2.3. Image Acquisition

MRI scans were performed on a 3.0T scanner (Achieva, Philips Healthcare, Amsterdam, The Netherlands) equipped with a dedicated eight-channel head coil. The imaging protocol always included coronal T1-weighted (repetition time (TR), 300 ms; echo time (TE), 16 ms; field of view (FOV), 150 mm; section thickness, 3 mm; matrix, 232 × 192), T2-weighted (TR, 300 ms; TE, 16 ms; FOV, 150 mm; section thickness, 3 mm; matrix, 232 × 196) and contrast-enhanced T1-weighted (TR, 300 ms; TE, 16 ms; FOV, 150 mm; section thickness, 3 mm; matrix, 216 × 167) sequence. Contrast-enhanced T1-weighted images were acquired 2 to 4 min after 0.1 mmol/Kg of the gadolinium contrast agent was fully injected at a rate of 2 mL/s via antecubital venous access.

### 2.4. Image Segmentation and Feature Extraction

MRI images were collected with the patient in a supine position. We confirmed the coronal layers of pre- and postoperative images, which included T1-weighted (T1WI), T2-weighted (T2WI) and contrast-enhanced T1 (T1CE). We selected the last image obtained before surgery as the preoperative image. The first postoperative image was obtained 3 months after surgery, which was selected for features extraction and statistical analysis. The images were preprocessed before feature extraction, including normalization, discretization and resampling to a 1 × 1 × 1 mm isotropic voxel size. The region of interest (ROI) was drawn manually by two researchers under the guidance of an experienced neurosurgeon and neuroradiologist on all patients’ T1WI, T2WI and T1CE scans using ITK-SNAP software (University of Pennsylvania, www.itksnap.org, accessed on 10 December 2021).

In total, 854 features were extracted from the segmented ROIs of each sequence by 3D-Slicer software. These features fell into four different categories: intensity histogram, texture, shape and wavelet; these contained eight types: first-order, gray-level cooccurrence matrix, gray-level dependence matrix, gray-level run length matrix, gray-level size zone matrix, neighboring gray tone difference matrix, shape and wavelet-based features. Wavelet transform is a mathematical method used to decompose signals or images into components with different frequencies. These frequency features are called wavelet features, which can help to describe the local changes and texture patterns in the image.

Feature selection and dimensionality reduction were mainly completed in R software (version 4.0.2). To evaluate quantitative feature consistency, we calculated the inter-observer correlation coefficient (ICC) and features with ICC values < 0.75 were excluded. Z-scores were used to complete data standardization, and all the features were normalized into a range of [−1, 1] [14]. Then, the minimum redundancy, maximum relevance method (mRMR) and the least absolute shrinkage and selection operator (LASSO) were performed to reduce dimensionality. The corresponding regularization coefficient (λ) was obtained by a 10-fold cross-validation in the LASSO regression based on the 1-standard error of the minimum criteria (1-SE criteria), and we confirmed the final screened image features.

### 2.5. The Establishment and Validation of a Radiomics–Clinical Model

The patients were divided into training and test sets at a ratio of 7:3 using random splitting. On T1WI, T2WI and T1CE, the pre- and postoperative radiomic features were combined and compared with the single pre- and postoperative radiomic features. We drew the receiver operator characteristic (ROC) curves and calculated area under curve (AUC) values to determine the postoperative image values. We build logistic regression models using pre- and postoperative features based on single (T1WI, T1CE and T2WI) and paired sequences (T1WI&T1CE, T1WI&T2WI and T1CE&T2WI) in combination with pre- and postoperative radiomic features. We estimated the predictive performance of these radiomic models using AUC analysis; this allowed us to determine the optimal sequence and generate an appropriate radiomic score (Rscore). A clinical model was established with meaningful indicators selected from clinical factors through univariate and multivariate analysis. Comparing the three models (Model 1: radiomics + clinical, Model 2: clinical, Model 3: radiomics), the one with the highest AUC value was considered the final model. The clinical and radiomic model was established in the training set and verified in the test set. A nomogram incorporating the radiomic signature and clinical risk factors was constructed to create an individual tool for the prediction of residual tumor regrowth. Calibration curves were plotted for the training and test sets, and the Hosmer–Lemeshow test was conducted to assess the agreement between the predicted risks and observed outcomes. A decision curve analysis (DCA) was performed to determine the clinical usefulness of the nomogram by quantifying the net benefits under different threshold probabilities. Figure 2 shows the analysis process of radiomics and nomogram construction.

### 2.6. Statistical Analysis

IBM SPSS Statistics version 26 (IBM SPSS Inc., Chicago, IL, USA) and R software (version 4.0.4) were used for all statistical analyses. Patients’ continuous variables are reported as means ± standard deviations or medians and interquartile range (IQR) as appropriate; categorical variables are presented as absolute and relative frequencies. We used the two-sample T-test to analyze normally distributed variables and Mann–Whitney’s test to analyze non-normally distributed variables. Categorical variables were compared using the Pearson chi-square test, continuity correction and Fisher exact test. A logistic regression model with multivariate analysis was performed to calculate the independent variables’ odds ratio (OR) and 95% confidence interval (CI). For all statistical analyses, *p* < 0.05 was considered statistically significant.

## 3. Results

### 3.1. Patients’ Characteristics

Patients’ baseline characteristics are shown in Appendix A. There were no significant between-group differences in clinical characteristics between the training and test cohorts. Of the 114 patients included in this study, 70 (61.4%) demonstrated residual tumor regrowth and 44 (38.6%) demonstrated no regrowth based on postoperative follow-up MRI images. The average follow-up time was 64.7 months (24–108 months). The medium TVDT was 1292.5 days (21–3010 days). Univariate analysis identified that headache (*p* = 0.009), Knosp classification (*p* < 0.001) and TVDT (*p* = 0.002) were risk factors for residual tumor regrowth (Table 1). Knosp classification [*p* = 0.001; OR = 2.255; (95% CI, 1.396–3.643)] and TVDT [*p* = 0.011; OR = 0.173; (95% CI, 0.045–0.669)] were ultimately confirmed as independent risk factors for the residual tumor regrowth by multivariate logistic regression. In contrast, headache [*p* = 0.14; OR = 2.515; (95% CI, 0.736–8.591)] was not a risk factor. In the patients with Knosp grades 3 and 4, the rate of residual tumor regrowth was 80.5% (29/36) and it was 45.5% (20/44) in patients with Knosp grades 0–2. The rate of residual tumor regrowth is 81.8% (27/33) in patients with TVDT < 1 year and 46.8% (22/47) in patients with TVDT ≥ 1 year. The patients with high Knosp classification (grades 3 and 4) and TVDT < 1 year had a higher regrowth rate of 93.3% (14/15).

### 3.2. Radiomic Feature Selection

We standardized 854 features of 6 images, and ICC was calculated to reduce feature extraction error. Thereafter, we retained features from preoperative T1WI (*n* = 529), preoperative T1CE (*n* = 401), preoperative T2WI (*n* = 567), postoperative T1WI (*n* = 523), postoperative T1CE (*n* = 468) and postoperative T2WI (*n* = 298). Then, mRMR and LASSO regression, performed in the R environment, were used to reduce dimensionality in the training set. The selected radiomic features are shown in Appendix A. In total, 4 were from the first-order features and 12 were from the second-order features (2 from gray-level co-occurrence matrix, 2 from gray-level run length matrix, 3 from gray-level size zone matrix, 5 from shape). We combined the pre- and postoperative radiomic features with single preoperative or postoperative radiomic features; the AUC values of a combination of pre- and postoperative radiomic features in T1WI, T2WI and T1CE were all higher (Appendix A). We build logistic regression models using pre- and postoperative features based on single (T1WI, T1CE and T2WI) and paired sequences (T1WI&T1CE, T1WI&T2WI and T1CE&T2WI), drew ROC curves, calculated AUC values and ultimately determined the optimal model. The T1WI&T1CE feature set was used to construct the radiomic model [AUC = 0.954, (95% CI, 0.909–0.999)] (Figure 3a,b). The features were substituted with a new Rscore using the following formula: Rscore = −0.2100 − 0.7805 × original_glcm_MCC − 0.7818 × wavelet.LHL_glrlm_Run Variance − 1.2187 × wavelet.LLL_glrlm_Long Run High Gray-Level Emphasis − 4.2583 × wavelet.HHL_glcm Cluster Shade + 0.1667 × wavelet.HHL_firstorder Skewness + 0.5335 × wavelet.LLL_firstorder Skewness − 0.8874 × original_shape Sphericity (extracted from postoperative T1CE).

### 3.3. Construction of the Radiomics–Clinical Model

Among the three significant clinical factors on univariate analysis, we excluded headaches by multivariate analysis. Incorporating the radiomic signature from the T1WI&T1CE MRI images, Knosp classification and tumor TVDT, three models were constructed by logistic regression. The models combined clinical risk factors and radiomic features (Model 1), single clinical factors (Model 2) and single radiomic features (Model 3). Model 1 was AUC 0.929 (95% CI, 0.865–0.993) and an accuracy of 0.888 for the training set; for the test set, the AUC was 0.882 (95% CI, 0.735–1.000) and an accuracy of 0.823. Model 2 was AUC 0.811 (95% CI, 0.704–0.918) and 0.762 for the training set; for the test set, the AUC was 0.834 (95% CI, 0.676–0.992) and an accuracy of 0.794. Model 3 was AUC 0.844 (95% CI, 0.748–0.941) and an accuracy of 0.787 for the training set; for the test set, the AUC was 0.763 (95% CI, 0.569–0.958) and an accuracy of 0.735. The radiomics–clinical model performed significantly better than the single clinical or radiomic model in both the training and test sets (Figure 3c,d, Table 2).

### 3.4. Individualized Nomogram Construction and Validation

The model that incorporated Rscore, Knosp classification and TVDT was developed and presented as a nomogram (Figure 4a). This nomogram showed good calibration in the training and test sets (*p* = 0.776 and 0.417, respectively) (Figure 4b). The DCAs for Models 1, 2 and 3 are shown in Figure 4c. The decision curve showed that if the threshold probability was >20%, then using the radiomics–clinical model to predict residual NF-PitNET regrowth added more benefit than using the single clinical or radiomic model in clinical application.

## 4. Discussion

As benign tumors mainly present with a mass effect, NF-PitNETs are principally resected using a transnasal trans-sphenoidal approach performed under endoscopic or microscopic visualization [3,5]. Unfortunately, only 40%–50% of surgical cases achieve complete resection [8]. Many studies have shown that patients with residual tumors have significantly higher rates of postoperative regrowth than patients who undergo total tumor resection [6,16,28]. However, the postoperative biological behavior of NF-PitNETs is variable. Some residual NF-PitNETs are stable over the long term, while others grow; previous studies showed that residual tumor regrowth or progression occurred in 47–64% of cases with partial resection [16]. Thus, we constructed a new model to predict postoperative residual tumor regrowth in patients with clinical NF-PitNET. Our results indicated that select radiomic features plus two readily available clinical variables (Knosp classification and TVDT) improved the predictive model’s performance. This model may help optimize individualized and stratified clinical treatment decisions.

Radiomic analysis can be applied to the whole tumor to obtain reproducible, objective and quantitative data from different imaging sequences to provide a more comprehensive approach to information acquisition [21]. Radiomics is increasingly applied to pituitary tumor research. Niu et al. [29] used radiomics to predict cavernous sinus invasion in NF-PitNET. Zhang et al. [18] used preoperative radiomic texture and histogram analysis to distinguish null cell adenomas from other subtypes in NF-PitNET. Zhang et al. [21] used radiomics to predict regrowth in non-functioning pituitary macroadenomas and radiomic analyses based on T1CE and T2WI in preoperative MRI. These studies suggest that radiomic features might be useful tools to predict the regrowth of residual NF-PitNET, but no reports regarding this concept have been published yet.

We first tried to compare the models based on pre- and postoperative images of T1WI, T1CE and T2WI to differentiate regrowth and non-regrowth groups. To our best knowledge, this is the first study to include postoperative radiomic features in its analysis. We found that combining pre- and postoperative images was better than using single pre- or postoperative images. This indicates that the postoperative images are significant referents for researching residual tumors. Next, we constructed a logistic regression based on single- (T1WI, T1CE and T2WI) and paired-sequence (T1WI&T1CE, T1WI&T2WI and T1CE&T2WI) models and drew a ROC curve to compare predictive performance. The results showed that T1WI&T1CE performed better than other images. This finding is consistent with many previous reports that affirmed the value of multi-modal imaging as superior to single-modal imaging for prognostication [30]. However, it is unknown if T1WI is better for discrimination and prediction of various aspects of PitNETs than T2WI [29]. The reason for this discrepancy may not be apparent, as the potential mechanism requires further study.

Two independent clinical predictors were included in the radiomics–clinical model for predicting postoperative residual tumor regrowth in patients with clinical NF-PitNET: Knosp classification and preoperetive TVDT. Tumors were considered invading the cavernous sinus if they demonstrated Knosp grades 3 or 4 [31]. The significance of Knosp classification is not particularly controversial in terms of tumour recurrence [4,7,31]. Our research shows that the Knosp classification significantly influenced residual tumor regrowth in univariate (*p* < 0.001) and multivariate (*p* = 0.001) analysis; a high Knosp classification (grades 3 and 4) indicated a higher risk of residual tumor regrowth after surgery; the rate of residual tumor regrowth was 80.5% in patients with Knosp grades 3 and 4, which is 45.5% in patients with Knosp grades 0–2. Preoperative TVDT also warrants consideration as a postoperative residual tumor regrowth risk factor. Miwa et al. [26] considered that preoperative tumor doubling time is useful in predicting the recurrence and malignancy of meningioma; their research concluded that preoperative tumor doubling time of less than 1 year have a significantly higher incidence of WHO grade 2, which means that preoperative TVDT may have significance in the malignancy and recurrence of intracranial tumors. A previous study showed that the postoperative TVDT was positively correlated to preoperative TVDT (r = 0.497, *p* = 0.026) in PitNETs [11], and this result suggested that the predictive ability of preoperative TVDT for recurrence of PitNET deserves further study. In our study, we found that preoperative TVDT was an independent risk factor in patients with postoperative residual tumors, and a short TVDT increased the risk of residual tumor regrowth; the rate of residual tumor regrowth was 81.8% in patients with TVDT < 1 year, which is 46.8% in patients with TVDT ≥ 1 year. The Ki-67 labeling index is generally considered a reliable marker of proliferation. Nuclear expression in >3% of tumor cells indicates a greater chance of regrowth [16,28]. A Ki-67 index greater than 3% is commonly used to highlight pituitary tumors with an unusually high proliferative index [15,16,32]. However, we found no significant correlation between ki67 and residual tumor regrowth. This may be due to an insufficient number of cases. Controversial factors noted by previous studies, such as age and surgical resection ratio [6,21,31,33], emerged as non-significant in our study.

In 2018, Matousek et al. [16] concluded that regrowth was more likely in adenomas with higher expression levels of the p21 cell cycle marker, the p53 tumor suppressor marker and a higher Ki-67 proliferative index. However, the researchers did not establish a model for identifying high-risk patients for stratified clinical management. In 2019, Cheng et al. [4] established a promising statistical model to predict NF-PitNET regrowth using clinical and protein signatures; however, the research was not entirely focused on patients with postoperative tumor residue. In addition, protein signature information is difficult to obtain in clinical practice settings. Our study used readily obtainable imaging and clinical information to establish a reliable risk stratification model that contributes to the prediction of postoperative residual tumor regrowth.

Our study has some limitations. First, this was a retrospective study from a single institution, and the sample size was not large. Further, well-powered studies with external validation are needed. Secondly, manual tumor ROI segmentation inevitably encounters irregularities, although two researchers jointly completed the image segmentation and ICC was calculated to reduce error. Thirdly, there is an increasing amount of multi-omics research; thus, radiomics can be combined with other omics, such as genomics, to more accurately identify tumors and guide comprehensive postoperative treatment.

## 5. Conclusions

We trained a novel radiomics–clinical predictive model for identifying patients with NF-PitNET at increased risk of postoperative residual tumor regrowth. This model could help optimize individualized and stratified clinical treatment decisions.

## Figures and Tables

**Figure 1 medicina-59-01525-f001:**
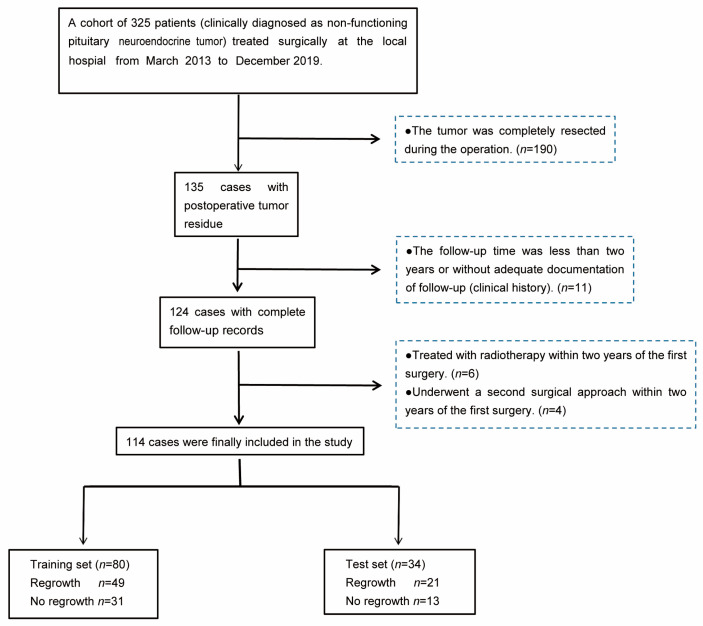
Flow diagram of patient enrollment.

**Figure 2 medicina-59-01525-f002:**
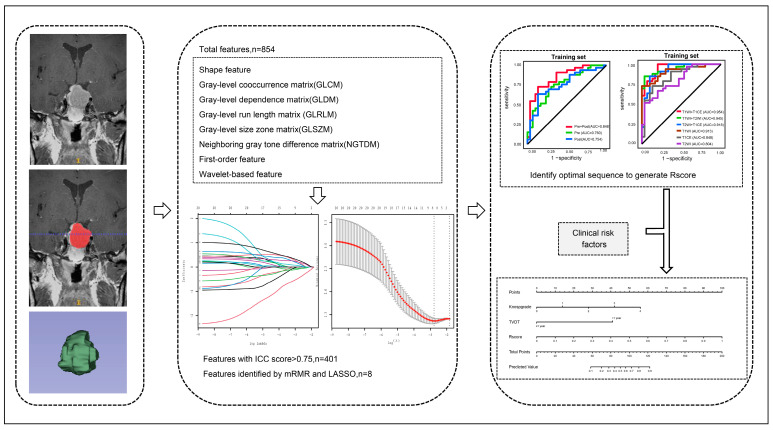
The analysis process of radiomics and nomogram construction. The radiomic feature extraction and dimension reduction process shown in the figure is from the preoperative coronal T1CE.

**Figure 3 medicina-59-01525-f003:**
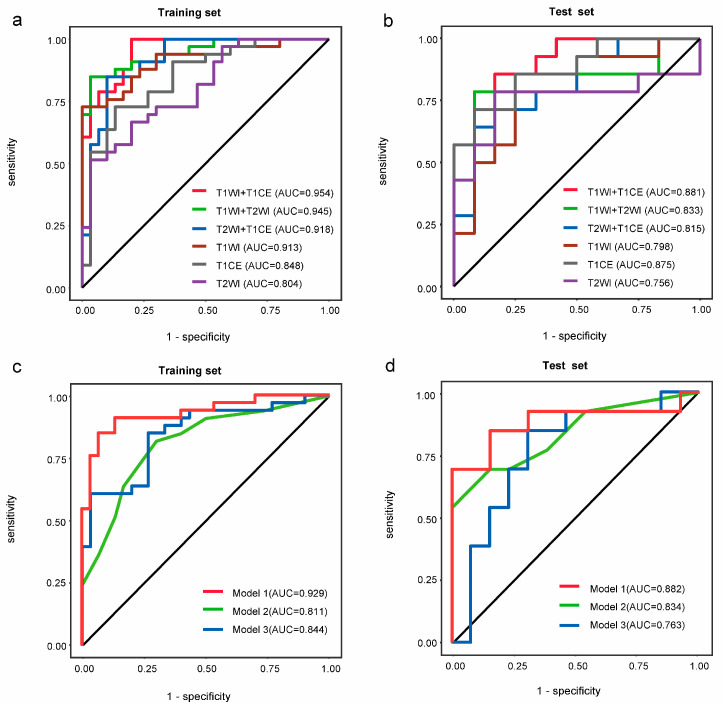
Construction of radiomics–clinical model. The ROC curve and AUC of single (T1WI, T1CE and T2WI) and paired sequences (T1WI&T1CE, T1WI&T2WI and T1CE&T2WI) in the training (**a**) and test sets (**b**), respectively. Each kind of images combined the pre- and postoperative radiomic features. The ROC curve and AUC of three models in the training (**c**) and test sets (**d**), respectively. Model 1 included clinical risk factors and radiomic features; Model 2 included single clinical risk factors and Model 3 included single radiomic features.

**Figure 4 medicina-59-01525-f004:**
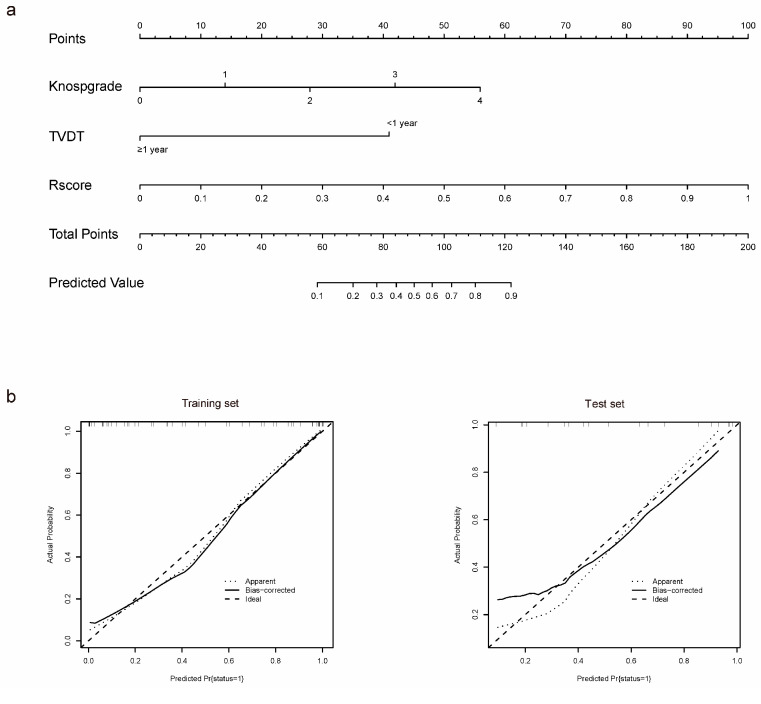
Construction and validation of the nomogram. (**a**) A radiomics–clinical nomogram incorporating the radiomic score, Knosp classification and TVDT on the training set. The predicted value presents the predictive probability of postoperative residual tumor regrowth in patients with clinical NF-PitNET. (**b**) Calibration curves of the radiomics–clinical nomogram in the training and test sets. The *x*-axis represents the nomogram-predicted probability and *y*-axis presents the actual probability of postoperative residual tumor regrowth. Ideal, perfect prediction corresponds to the 45° line. Apparent, entire cohort. Bias-corrected, the line bias-corrected by bootstrapping (B = 100 repetitions), indicated observed nomogram performance. (**c**) Decision curve analysis for the radiomics–clinical model, radiomic model and clinical model. The decision curve showed that if the threshold probability was >20%, then using the radiomics–clinical model to predict residual NF-PitNET regrowth added more benefit than using the single clinical or radiomic model in clinical application. Clin + radio, radiomics–clinical model; Radio, radiomic model; Clin, clinical model.

**Table 1 medicina-59-01525-t001:** Clinical characteristics of patients in the training and test sets.

Variables (*N* = 114)		Training Set (*N* = 80)			Test Set (*N* = 34)	
	Residual	No Residual Regrowth (*N* = 31)	*p*-Value	Residual	No Residual Regrowth	*p*-Value
	Regrowth (*N* = 49)			Regrowth (*N* = 21)	(*N* = 13)	
Gender, *n* (%)			0.935			0.481
Male	28 (57.1%)	18 (58.1%)		12 (57.1%)	9 (69.2%)	
Female	21 (42.9%)	13 (41.9%)		9 (42.9%)	4 (30.8%)	
Age (years)	54 (41.5–66)	51 (41–66)	0.771	51 (44–70)	46 (36–61)	0.103
Weight (kg)	64.0 (59.0–71.5)	64.0 (58.0–69.0)	0.909	60.0 (53.0–65.0)	62.0 (58.0–72.0)	0.166
Height (m)	1.65 ± 0.07	1.62 ± 0.06	0.103	1.63 ± 0.08	1.65 ± 0.07	0.363
BMI	23.66 (21.08–26.34)	23.95 (22.04–26.57)	0.295	22.04 (21.22–23.95)	22.17 (20.00–25.96)	0.818
Headache, *n* (%)			0.009 *			0.004 *
Yes	29 (59.2%)	9 (29.0%)		14 (66.7%)	2 (15.4%)	
No	20 (40.8%)	22 (71.0%)		7 (33.3%)	11 (84.6%)	
Vision changes, *n* (%)			0.444			0.727
Yes	28 (56.2%)	15 (48.4%)		13 (61.9%)	9 (69.2%)	
No	21 (43.8%)	16 (51.6%)		8 (38.1%)	4 (30.8%)	
Pituitary apoplexy, *n* (%)			0.931			0.498
Yes	6 (12.2%)	4 (12.9%)		2 (9.5%)	2 (15.4%)	
No	43 (87.8%)	27 (87.1%)		19 (90.5%)	11 (84.6%)	
Knosp grade, *n* (%)			<0.001 *			0.005 *
0	2 (4.1%)	12 (38.7%)		1 (4.8%)	7 (53.8%)	
1	6 (12.2%)	8 (25.8%)		3 (14.3%)	2 (15.4%)	
2	11 (22.4%)	5 (16.1%)		2 (9.5%)	2 (15.4%)	
3	11 (22.4%)	4 (12.9%)		5 (23.8%)	1 (7.7%)	
4	19 (38.9%)	2 (6.5%)		10 (47.6%)	1 (7.7%)	
Cystic, *n* (%)			0.433			0.427
Yes	15 (30.6%)	7 (22.6%)		7 (33.3%)	2 (15.4%)	
No	34 (69.4%)	24 (77.4%)		14 (66.7%)	11 (84.6%)	
Hardy grade, *n* (%)			0.54			0.191
0	16 (32.6%)	6 (19.4%)		4 (19.0%)	1 (7.7%)	
I	19 (38.8%)	14 (45.2%)		8 (38.1%)	4 (30.8%)	
II	10 (20.4%)	9 (29.0%)		4 (19.0%)	6 (46.2%)	
III	2 (4.1%)	0 (0%)		4 (19.0%)	0 (0%)	
IV	2 (4.1%)	2 (6.4%)		1 (4.9%)	2 (15.3%)	
TVDT, *n* (%)			0.002 *			<0.001 *
<1 year	27 (55.1%)	6 (19.4%)		15 (71.4%)	1 (7.7%)	
≥1 year	22 (44.9%)	25 (80.6%)		6 (28.6%)	12 (92.3%)	
Surgical resection ratio (%)	67.34 (16.25–98.58)	69.68 (21.69–98.60)	0.933	73.64 (37.10–98.01)	80.81 (22.54–95.59)	0.326
Consistency, *n* (%)			0.083			0.574
Soft	28 (57.1%)	21 (67.7%)		14 (66.7%)	11 (84.6%)	
Medium	7 (14.3%)	0 (0%)		2 (9.5%)	0 (0%)	
Hard	14 (28.6%)	10 (32.3%)		5 (23.8%)	2 (15.4%)	
Residual position, *n* (%)			0.349			0.894
Intrasellar	20 (40.8%)	15 (48.4%)		10 (47.6%)	5 (38.5%)	
Suprasellar	16 (32.7%)	12 (38.7%)		8 (38.1%)	5 (38.5%)	
Both	13 (26.5%)	4 (12.9%)		3 (14.3%)	3 (23.0%)	
Postoperative T1 enhancement, *n* (%)			0.029			0.21
Yes	34 (69.4%)	28 (90.3%)		15 (71.4%)	12 (92.3%)	
No	15 (30.6%)	3 (9.7%)		6 (28.6%)	1 (7.7%)	
Ki-67, *n* (%)			0.132			0.21
<3%	30 (61.2%)	24 (77.4%)		15 (71.4%)	12 (92.3%)	
≥3%	19 (38.8%)	7 (22.6%)		6 (28.6%)	1 (7.7%)	

* Represents the number of *p*-values < 0.05. Abbreviation: TVDT, Tumor volume doubling time.

**Table 2 medicina-59-01525-t002:** Predictive performance of Models 1, 2 and 3 in the training and test sets.

	Training Set (*n* = 80)				Test Set (*n* = 34)			
	AUC (95% CI)	ACC	SEN	SPE	AUC (95% CI)	ACC	SEN	SPE
Model 1	0.929 (0.865, 0.993)	0.888	0.848	0.933	0.882 (0.735, 1.000)	0.823	0.818	0.833
Model 2	0.811 (0.704, 0.918)	0.762	0.818	0.700	0.834 (0.676, 0.992)	0.794	0.800	0.785
Model 3	0.844 (0.748, 0.941)	0.787	0.848	0.733	0.763 (0.569, 0.958)	0.735	0.764	0.705

Model 1 included both clinical risk factors and radiomic features, Model 2 included single clinical risk factors and Model 3 included single radiomic features. AUC, area under the curve; ACC, accuracy; CI, confidence interval; SEN, sensitivity; SPE, specificity.

## Data Availability

All data analyzed during this study are included in this article. Further inquiries can be directed to the corresponding author.

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
