# Peer review of "A Novel Magnetic Resonance Imaging-Based Radiomics and Clinical Predictive Model for the Regrowth of Postoperative Residual Tumor in Non-Functioning Pituitary Neuroendocrine Tumor"

_medicina, 2023, doi:10.3390/medicina59091525_

Round 1

Reviewer 1 Report

Reviewer

Initial comments

      This paper is important, but needs to be corrected, as there are conceptual errors, as already noted in the title and also in the introduction and sequence of the work.

Review

Abbreviations – The paper has many acronyms, and a list of abbreviations would be advisable for a better understanding of the text.

Title:

A novel MRI-based radiomics and clinical predictive model for the regrowth of postoperative residual tumor in clinically non- functioning pituitary neuroendocrine tumor

Comment:

Lines 2,3,4……

A novel MRI-based……

 ....magnetic resonance imaging......

MRI-….. Please put in full

….in clinically non- functioning pituitary neuroendocrine tumor….

Please remove the word clinically and neuroendocrine.

And wouldn't it be better to put - non-secretory pituitary tumor as a macroadenoma?

Because according to the size of the pituitary tumor (such as non-secretory microadenomas), we may not have a clinic.

Abstract:

Comment:

 Line 9- Background and Objectives:

Lines 10,11….. in the patients with clinically non-functioning pituitary neuroendocrine tumors (NF-PitNETs).

Please, are these tumors macroadenomas or microadenomas?

Which size?

 Lines – 25, 26,27,28…..Conclusions:  We trained a novel  radiomics-clinical predictive model for identifying patients with NF-PitNETs at increased risk of  postoperative residual tumor regrowth. This model may help optimize individualized and stratified clinical treatment decisions.

Please, what would the clinical treatment be like?

1. Introduction  

 Comment:

 Lines 31,32,33….Introduction  

Pituitary neuroendocrine tumors (PitNETs) constitute 10%–20% of all primary brain tumors [1,2].

Reference 1- Previous studies attempting to define the natural history of postoperative nonfunctioning pituitary adenomas (pNFPAs)

Please name the pituitary tumor...... nonfunctioning pituitary adenomas

To correct……. Pituitary neuroendocrine tumors……

 Lines 33,34,35,36…..Clinically non-functioning pituitary neuroendocrine tumors (NF-PitNETs)  account for 14%–54% of PitNETs. These tumors are often associated with mass symptoms such as visual impairment, headache, and varying degrees of hypopituitarism without abnormal increases in serum hormone levels [3,4].

Please, what size are these pituitary tumors to refer these clinical symptoms?

2. Materials and Methods

2.1. Patient selection

Comment:

It is suitable

2.2. Clinical characteristics and definitions

Comment:

Line 92….BMI (calculated by height and weight),

Please put the correct one....

(BMI = weight (kg) / height ² (m²)),

Line 93…Knosp classification

Line 94... Hardy classification,...

Please, what about the references?

2.3. Image acquisition

Comment:

It is suitable

2.4. Image segmentation and feature extraction

Comment:

It is suitable

2.5. The establishment and validation of a radiomics-clinical model

Comment:

It is suitable

2.6. Statistical analysis

Comment:

It is suitable

3. Results  

3.1. Patients’ characteristics

Comment:

It is suitable

3.2. Radiomics feature selection

Comment:

It is suitable

3.3. Construction of the radiomics-clinical model

Comment:

It is suitable

3.4. Individualized Nomogram construction and validation

Comment:

It is suitable

4. Discussion

Comment:

It is suitable

5. Conclusions

Comment:

Lines 373, 374, 375, 376….Conclusions 

         We trained a novel radiomics-clinical predictive model for identifying patients with NF-PitNET at increased risk of postoperative residual tumor regrowth. This model could help optimize individualized and stratified clinical treatment decisions.

Please, what would the clinical treatment be like?

Supplementary Materials:

Comment:

It is suitable

References

Comment:

It is suitable

But the references are missing.

Line 93…Knosp classification

Line 94... Hardy classification,...

Thank you

Reviewer 2 Report

Authors aim to develop a predictive model for NF-PitNET recurrence based on radiomics features and clinical assesment as observed in a single center retrospective cohort of patients. This is an overall interesting approach, and the results are significant. Major shortcomings of the study are adequately discussed.

A few suggestions here, as to the improvement of the manuscript:

Description and illustration of the selected radiomics features, rather than only tabulating them, may help the reader better understand the approach that was chosen here.

How do authors explain that Knosp grading appears to be a better predictor of tumor recurrence than Hardy grading?

Minor spell check advised.

Reviewer 3 Report

The submitted manuscript titled " A novel MRI-based radiomics and clinical predictive model for the regrowth of postoperative residual tumor in clinically non-functioning pituitary neuroendocrine tumor " by Shen et al. investigates predictive factors of tumor regrowth in non-functioning pituitary adenomas (NF-PitNET) using clinical and radiological data (the latter in the pre- and post-operative setting). A thorough analysis of diverse radiological parameters ("radiomics") in different models were investigated for their usefulness in this particular context and forming a Rscore. As was to be expected from prior research, Knosp grade and tumor growth rate (expressed as tumor volume doubling time) were predictors of residual regrowth. Whereas the radiomics model is quite complicated and can't be pinpointed to single features.  

Overall, I am impressed with the quality of the article and believe that it makes a valuable contribution to the literature on this topic. In particular, I want to praise the clear structure of the article, the thorough and laborous data collection and analysis, and the overall quality of presentation.

The article is well-structured, with clear headings and subheadings that guide the reader through the content. In terms of language, the article is well-written and easy to follow. The authors use appropriate scientific terminology and provide clear definitions when necessary. The article is also free of spelling and syntax errors, which is a credit to the authors' attention to detail.

One possible weakness of the article is that it doesn't seem to be generalizable easily. The radiomics model seems quite specific and only further research will tell whether or not these parameters derived from other manufacturers or MRI scanners with lower resolution may predict residual regrowth equally well.

In conclusion, the manuscript at hand is a well-written and thoroughly researched article that provides valuable insights into the topic and offers an interesting and timely approach to assess prognosis in such patients.
